# River thorium concentrations can record bedrock fracture processes including some triggered by distant seismic events

Benjamin Gilbert [1,2] ✉, Sergio Carrero [1,3], Wenming Dong [1], Claresta Joe-Wong [1], Bhavna Arora [1], Patricia Fox [1], Peter Nico [1] & Kenneth H. Williams [1,4]

Fractures are integral to the hydrology and geochemistry of watersheds, but our understanding of fracture dynamics is very limited because of the challenge of monitoring the subsurface. Here we provide evidence that long-term, high-frequency measurements of the river concentration of the ultra-trace element thorium (Th) can provide a signature of bedrock fracture processes spanning neighboring watersheds in Colorado. River Th concentrations show abrupt (subdaily) excursions and biexponential decay with approximately 1-day and 1-week time constants, concentration patterns that are distinct from all other solutes except beryllium and arsenic. The patterns are uncorrelated with daily precipitation records or seasonal trends in atmospheric deposition. Groundwater Th analyses are consistent with bedrock release and dilution upon mixing with river water. Most Th excursions have no seismic signatures that are detectable 50 km from the site, suggesting the Th concentrations can reveal aseismic fracture or fault events. We find, however, a weak statistical correlation between Th and seismic motion caused by distant earthquakes, possibly the first chemical signature of dynamic earthquake triggering, a phenomenon previously identified only through geophysical methods.

Fractures in bedrock at the base of the critical zone enable geogenic elements to enter rivers and biogeochemical cycles in catchments and downstream hydrologic systems[1]. Fractures expose rock surfaces to groundwater, initiating the chemical weathering reactions[2] that ultimately transform protolith to soil, and provide the fluid flow pathways that connect groundwater with surface waters. Borehole records[3,4] and active geophysical measurements[5,6] are increasingly used to investigate the fractures distributions in watersheds but are generally limited to observing static structures. Passive seismic monitoring is sensitive to motions on faults, including earthquakes that can change flow pathways, permeability and groundwater chemistry[7–10]. Surficial seismic studies have captured rock fracturing associated with stress and weathering[11], and provided new insights into environmental controls on rock failure[12]. However, there are few methods and studies that detect any changes to near-surface bedrock fracture.

Here we report that transient thorium concentrations obtained from in long-term, high-frequency water chemistry monitoring can provide a chemical signature for subsurface fault or fracture events, likely affecting fault-zone water and solute transport, in mountainous shale-dominated watersheds.

## Results

### River water chemistry

The monitoring program is part of the Watershed Function Science Focus Area project that has established a community field observatory within the greater East River watershed, CO, with satellite

[1]Energy Geoscience Division, Lawrence Berkeley National Laboratory, 1 Cyclotron Road, Berkeley, CA 94720, USA. [2]Department of Earth and Planetary Science, University of California, Berkeley, CA 94720, USA. [3]Institute of Environmental Assessment and Water Research, CSIC, Jordi Girona 18, 08034 Barcelona, Spain. [4]Rocky Mountain Biological Laboratory, Gothic, CO 81224, USA. ✉e-mail: bgilbert@lbl.gov

measurement sites in neighboring catchments including the main stem East River and Coal Creek (Fig. 1A)[13]. Watersheds in this region are situated in late Cretaceous sedimentary rock altered by episodes of Cenozoic igneous activity. Coal Creek experienced more intense metamorphism and accompanying sulfide mineralization than the East River, leading to historic mining activities and enhanced river metal concentrations[14]. Because time-dependent measurements of river chemistry and discharge provide distinctive signatures of watershed processes[15,16], an extensive suite of elements and species are monitored frequently (up to daily) at the Pump House (PH) location of the East River and the Coal-11 location of Coal Creek, and at less frequent sampling intervals at additional locations throughout the watersheds (Figs. 1 and S1–3). The data show numerous patterns in major and trace element concentration that are consistent with seasonal trends in mineral reaction and biogeochemical cycling. For example, variation in the concentrations of base cations, dissolved inorganic carbon and sulfate at PH is consistent with swings between pyrite and carbonate weathering associated with water-table excursions through the water year[17,18] (Fig. S2). River metal(loid) concentrations are also consistent with release from sulfide weathering.

In contrast, the concentrations of thorium in both the East River and Coal Creek exhibited unusual but reproducible dynamics, characterized by a sudden increase above background followed by a decay (Fig. 1B). Twenty-two such episodes were recorded at the PH over 20 months in calendar years 2016–2018. The transients in thorium concentration are fitted by a biexponential with fast and slow exponential decays (Fig. S4) with time constants that ranged from 0.38–1.6 days and 1.8–8.2 days, respectively (Table S1). For the largest excursions in each watershed the transients are detectable above background for up to 3 weeks. Two further elements, arsenic and beryllium, exhibited simultaneous excursions with thorium in the East River (Fig. 1C; Fig. S5). No correlations between thorium and arsenic, beryllium or any other elements were observable at the Coal Creek, likely because the background concentrations of these elements, and their fluctuations, were much greater in the mining affected watershed

than in East River. Thorium concentrations at other measurement sites in the East River and Coal Creek watersheds showed similar behavior (Fig. S6). Some thorium excursions were temporally correlated between both hydrologically connected and unconnected measurement locations, evidence of a geographically dispersed phenomenon.

Thorium inputs into river water could originate from bedrock, soil or vegetation in the watershed or from the atmosphere, and we used chemical, meteorological and seismic data to test hypotheses for origins and input pathways. For different scenarios we used Event Coincidence Analysis[19] (ECA) to test the null hypothesis that thorium excursions and any other event type, such as precipitation, are random and uncorrelated phenomena. For sufficiently sparse events, analytical expressions (Eqs. S1 and S2) have been derived for the probability of $N_C$ coincidences where an event from a series A of size $N_A$ precedes an event from series B of size $N_B$, and for the p-value of observing $N_C$. ECA anlaysis was performed parametrically, varying the threshold for precursor event identification and the time window from 1 – 2.5 days. Thorium data from East River and Coal Creek were combined for ESA analyses and the criterion for thorium events was fixed at 0.05 μM increase between successive days.

## Test for thorium correlations with precipitation and other events

We first explored whether precipitation events were correlated with Th excursions using data from two meterological stations shown in Fig. 1A. Precipitation could cause increases in solute levels because large rainfall events can rapidly saturate soils and the vadose zone, flushing solutes into groundwater. Precipitation can also cause wet deposition of atmospheric aerosols, such as combustion products from coal-burning powerplants operating in Colorado and farther upwind in the Upper Colorado River Basin. Over this period, for example, USGS records of mercury deposition at the nearest atmospheric deposition measurement site, show a correlation with precipitation events (Figs. S3A and S3C). The parametric ECA test, however, did not find any statistically support for thorium excursions

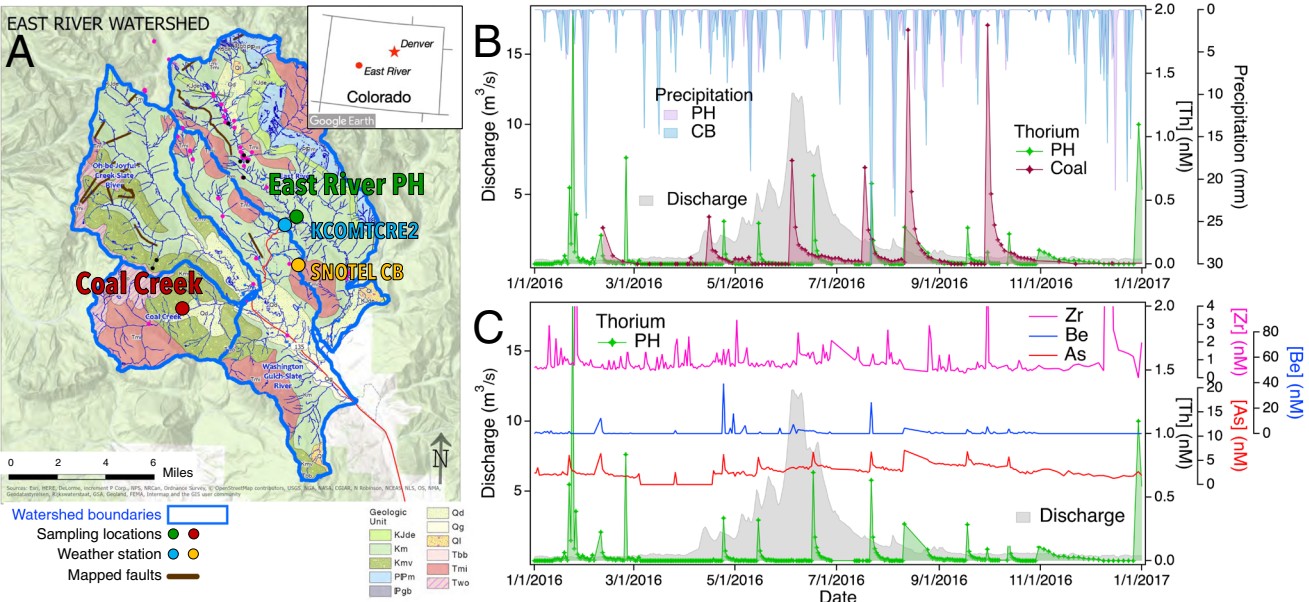

**Fig. 1 | Field site and measurements of river water chemistry. A** Geologic map of the East River and Coal Creek watershed with locations of principal river water sampling sites (red and green circles), weather stations (blue and yellow circles), and mapped faults (black lines). Map was generated using ArcGIS (Esri, Inc.) with hydrologic data from the National Hydrography Dataset and geologic information from the Gunnison County Geologic Map. See Code Availability Statement for full

credits and references. Inset: Map showing location of East River, adapted from Google Earth. **B** Comparison of East River Pump House (PH) and Coal Creek thorium data in calendar year 2016 with East River discharge and precipitation at PH and Crested Butte (CB). See Fig. S1 for 2017. **C** Comparison of East River thorium concentrations with zirconium, beryllium and arsenic. See Fig. S1 for 2017. See Figs. S2–3 for other data.

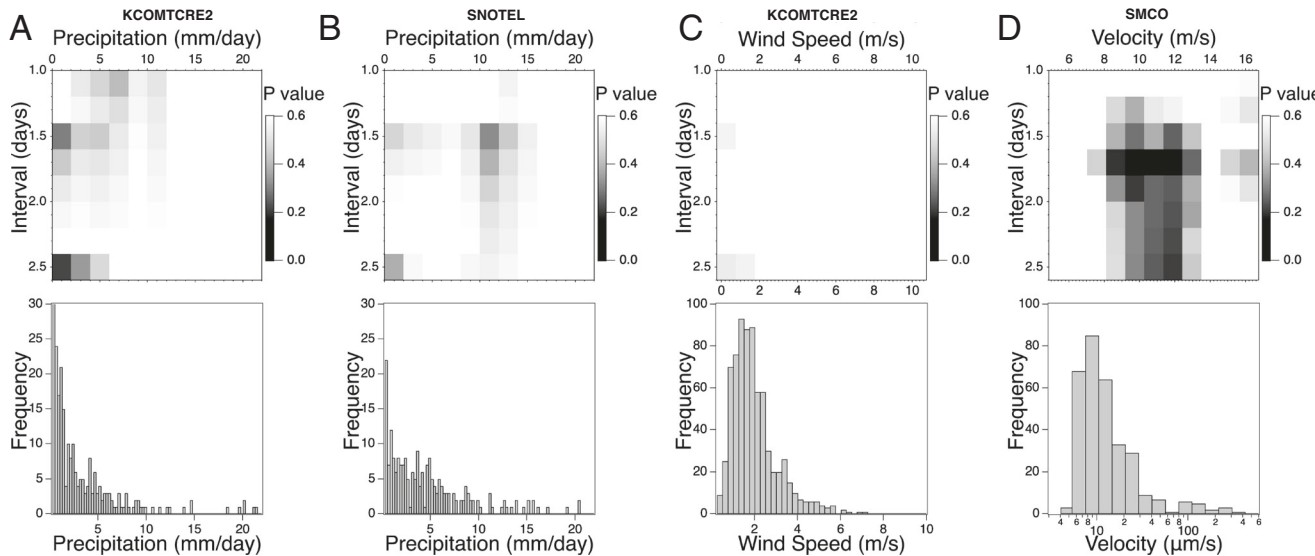

**Fig. 2 | Parametric event coincidence analysis (ECA) for daily precipitation, wind and seismicity events preceding thorium excursion (top row) and histograms of daily precipitation totals, wind speed and ground motion (bottom row). A** Precipitation data from KCOMTCRE2 weather station (location mapped in Fig. 1A). **B** Precipitation data from Crested Butte SNOTEL weather station (Fig. 1A). **C** Wind speed data from KCOMTCRE2 (Fig. 3A). **D** Seismicity data from SMCO (Fig. 3A).

being associated with precipitation, with the threshold for precipitation events varied between 0.5–30 mm per day, and the time window varied from 1–2.5 days (Fig. 2A). Moreover, solutes known to exhibit flushing behavior show no visual association with the thorium data (Fig. S2A, D).

Dry deposition of mineral dust originating from the Colorado Plateau and transported by winter storms can significantly affect the albedo of snow in the San Juan mountains in southwestern Co[20]. However, the seasonality and frequencies of the recorded dust-on-snow events[21] do not match East River thorium observations. Moreover, element leaching studies of mineral dust collected in the San Juan mountains found that thorium was released only under strongly acidic conditions not observed in the East River[22]. Furthermore, while copper and mercury accumulate on the soil surface due to atmospheric deposition, thorium is depleted at the surface relative to bedrock (Fig. S7).

Both catchments are underlain by sedimentary rock that could release thorium through hydrologically driven weathering. For example, the East River is situated in Mancos Shale that contains 4–10 ppm Th. However, time-series concentration data for major anions and cations released into groundwater by shale bedrock weathering (e.g., sulfate, calcium) do not show any abrupt concentration changes that are simultaneous with thorium (Fig. S2B, E) and the detrended $SO_4^{2-}$ and Ca data are not correlated with Th (Fig. S5C). Thus, hydrologically-driven weathering trends, in which water table variation is a dominant influence on solute export[17,23], do not cause abrupt thorium excursions, suggesting a geomechanical cause.

The East River watershed experiences occasional landslides that can transport partially weathered rock and soil into the river and that may enhance weathering[24]. A documented landslide in the East River in July 2017[25] coincided with an instrument failure that prevented the collection of water samples. Although ambient seismic methods can identify landslides[26], this capability is not installed at the site. However, that episode showed that landslides are associated with large changes in river water turbidity. The turbidity data, however, shows only 3 excursions that could be correlated with thorium (Fig. S8). Tree roots can stress and fracture rocks when trees are subjected to high winds[27], but average and maximum wind speeds do not visually correlate with thorium excursions (Fig. S2C, F) and ECA analysis finds no *p*-value below 0.41 (Fig. 2C).

A further hypothesis for the sudden introduction of thorium and other elements into the river is the creation or displacement of a fault or fracture that could expose minerals to undersaturated groundwater or open flow pathways[3]. X-ray tomography of a core drilled close to PH clearly show that shale weathering initiates at fracture surfaces (Fig. S9). To investigate the potential release of thorium from fresh surfaces in unweathered and partially-weathered Mancos shale from four East River locations (Fig. S10) was investigated by exposing powdered shale to simulated river water under aerobic (surficial) and anaerobic (subsurface) conditions for 48 h. Thorium was released in all cases, with the highest concentration from the deepest core and in anaerobic rather than aerobic solutions (Table S2). Arsenic was also released although beryllium was not detected.

## Groundwater chemistry

Field evidence for a subsurface bedrock origin of thorium is provided by depth-resolved groundwater concentrations acquired from monitoring wells along a hillslope transect[23,28], acquired in May and June 2017, a period when storm activity does not typically cause atmospheric dust transport and dry deposition. The field study observed a transient thorium excursion (accompanied by arsenic) localized at depth (e.g., at 3.7 m at PLM 3) with no corresponding near-surface concentration (Fig. 3). The maximum groundwater thorium concentration (1.8 ppb, equal to 7.8 nM) is significantly higher than observed PH concentrations during this period (0.07 ppb) and throughout this 20-month period (0.95 ppb). Later in the same field season, studies of shallow groundwater chemistry from a series of piezometers in the East River floodplain, very close to the Pumop House, showed lower thorium concentrations that are not correlated with East River values (Fig. 3C). This comparison further suggests a subsurface rather than atmospheric Th source.

## Thorium correlations with seismicity

To investigate whether fracturing processes could be responsible for thorium excursions in the field, we investigated local seismic data. Four seismic stations on the Intermountain West network were active in this period and region (Fig. 4A) with Snowmass (SMCO) the closest at ~50 km. The seismic data (Fig. 4B and Fig. S11) contain lower-intensity records of ground motion, as well as higher-intensity traces that could mostly be attributed to catalogued seismic events in

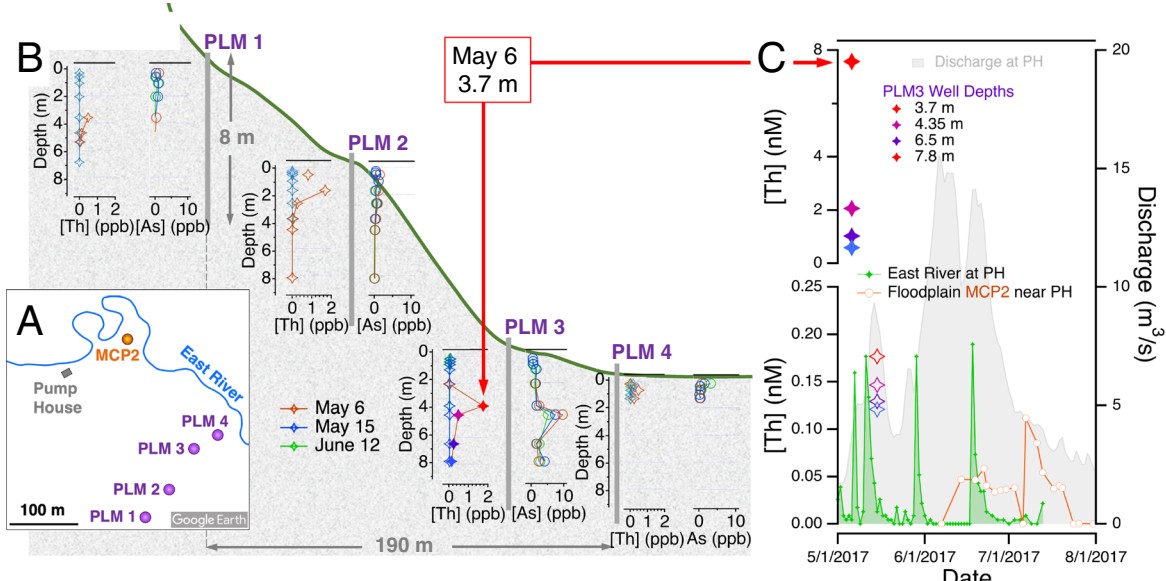

**Fig. 3 | Coupled thorium and arsenic excursions observed in deeper ground-water (>3 m) but not shallower groundwater in the hillslope or floodplain.** **A** Aerial view of Pump House on the East River with locations of monitoring wells PLM1–4 and meander piezometer MCP2 indicated. **B** Depth-resolved thorium and arsenic ground water concentrations measured in PLM wells on 3 days in May and June 2017. Plots are positioned on cross-section through hillslope topography (vertical scale enhanced). **C** Comparison of thorium data from deeper groundwater (PLM3), from shallower groundwater in the floodplain (MCP2) and East River. Note that two vertical axis scales are used to display ground and river thorium concentrations. Red arrows in (**B**) and (**C**) indicate the same thorium data point.

California or Nevada (Table S3). Magnitudes of these events are 2.6 or lower, far smaller than the larger earthquakes (M > 5) in Japan, Iceland, China, and Italy that caused detectable groundwater changes[9,10,29,30], and the maximum SMCO ground velocity is ~175 μm/s.

We performed a parametric ECA to test whether seismic motion preceded Th excursions more frequently than expected for random events. The analysis revealed a range of ground-motion threshold values and time-window durations for which the *p*-values were between 0.12–0.25 (Fig. 2D). Although the lowest *p*-value obtained in this analysis ($p = 0.12$) does not reach the threshold to be considered statistically significant ($p = 0.05$), these values are lower than attained for either precipitation data set. Moreover, the parametric ECA shows a cluster of low *p*-values that is not present in either of the precipitation data. Thus, suprisingly, ECA suggests that a portion of these bedrock fracture processes could be triggered by small ground motions caused by distant earthquakes. The initiation of local seismic events by propagating seismic waves, called dynamic earthquake triggering, is well established[7,31,32]. Although the ground motions detected here are at the lower end of the velocities that have been attributed to dynamic triggering, no lower bound in motions and stresses for this phenomenon has been established[32].

## Discussion

The above observations provide evidence that abrupt bedrock fracture processes can generate detectable trace solute inputs to ground and stream water. These could include processes that create new water-rock interactions, such as fracture creation or slip, as well as processes that change subsurface hydraulic pathways. The bedrock fracture processes inferred to mobilize thorium are likely to be partly caused by topographic (gravitational) and tectonic stresses[33], including stresses associated with cyclic ~2-cm annual variations in elevation observed by GPS monitoring throughout the Rockies (Fig. S3B). Detrended elevation values, however, do not show any correlation with East River thorium (Fig. 3C). Because the solutes exhibit fast increases in neighboring watersheds, we infer a role for stream- and river-spanning fault zones, a few of which have been mapped (Fig. 1A). Deep bedrock fracture structure plays important pathway for lowland

recharge from mountainous catchments and can facilitate inter-basin flow[34].

Thorium speciation and the cause of thorium mobility are not well constrained in this study. $Th^{4+}$ ions are poorly soluble in water, and readily sorb to silicate surfaces[35], but solubility is greatly enhanced by complexation with common oxyanions[36]. In particular, sulfate is a strong ligand for thorium that, due to sulfidic rock weathering in both watersheds, is present at orders-of-magnitude higher concentrations than, and thus could solubilize, thorium. However, abrupt changes in daily element concentrations are frequently associated with particle-associated transport. Working at the Catalina-Jemez Critical Zone Observatory, Olshansky *et al.* compared the elemental composition of the colloidal fraction and filtered stream water[37]. Those results showed that that Fe, Zr, Mn, Ti and Al were present in particulate phase and that particulate transport was associated with very high variance in concentration in time series data. In the East River data, those elements, as well as Th, Be and As, and others, show high-variance time-series patterns that could be consistent with particulate transport. Time-series and element-correlation plots show that East River Th data are not associated with Zr (Fig. 1C and Fig. S4B), Mn, Ti or Al, and those elements do not show exponential decay behavior. In contrast, a fraction of Be and As measurements are correlated with Th (Fig. S4B). More studies will be needed to establish if thorium excursions involve solute or fine-grained particulate concentrations.

We sought to test whether Th or other chemical signatures of watershed fault dynamics could reside in high-frequency river water data from other field sites. However, the identification of the thorium signal in this study was enabled by a long-term water monitoring program in which measurement interval (1–2 days) was smaller than the characteristic timescale for the thorium excursions[38]. and we found only two data sets to compare with the present study (Fig. S11). Daily thorium measurements from Kervidy-Naizin watershed show entirely different behavior for an intensively farmed, low topography landscape set on schist[39]. Time-series and correlation plots between detrended Th and Na data show that both solutes at Kervidy-Naizin are affected by hydrologic variation (Fig. S11). It is plausible that mountainous watersheds, which are often characterized by a greater

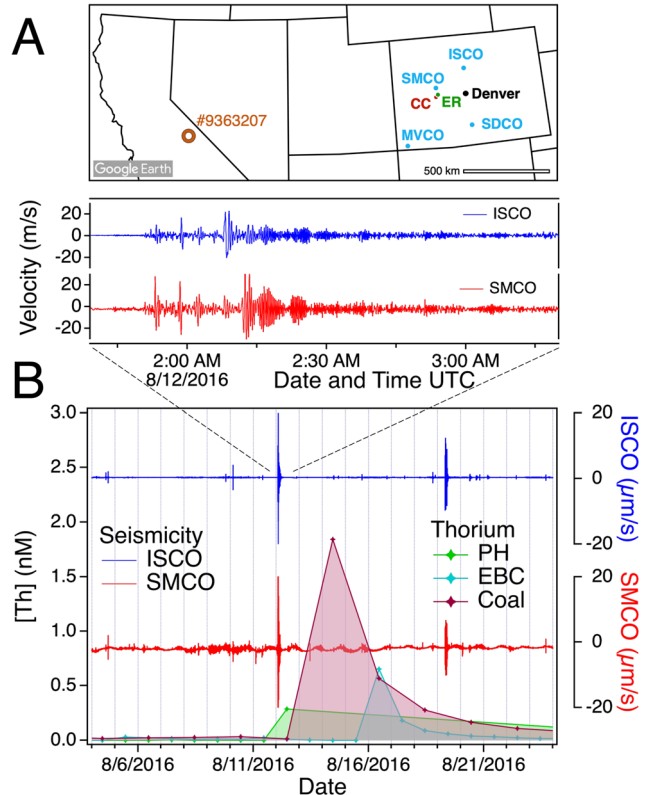

**Fig. 4 | Seismicity and thorium concentration. A** Map of the western US showing the location of East River (ER) and Coal Creek (CC) in Colorado, seismic monitoring stations and the epicenter of seismic event #9363207. Map adapted from Google Earth. **B** Example of seismicity preceding thorium excursions in ER and CC that was recorded at two stations in Colorado and attributed to catalogued event #9363207.

role for bedrock fractures compared to overlying soil in water storage and transport, may be most favorable for identifying geochemical signatures of fracture processes. Long-term river water measurements from the geologically more similar Krycklan watershed set in a metasedimentary basin[40] are too low-frequency to permit comparison.

The thorium signal thus provides a window into bedrock mechanical processes that presently evade geophysical detection. Future studies of local seismicity and water chemistry are envisioned that could test hypotheses for deep rock-water interactions controlling critical-zone processes[1].

## Methods

### Stream and groundwater sampling
The collected stream and porewater samples were filtered in the field (0.45 μm polytetrafluoroethylene syringe filters), and divided into subsamples for different types of analyses. The subsamples for cation analysis were immediately acidified into a 2% nitric acid ($HNO_3$) matrix by adding concentrated ultrapure $HNO_3$. The samples were shipped overnight to the laboratory in cooler containing ice packs, and stored in a 3 °C refrigerator for later analyses.

### Analytical methods
The concentrations of Th, Zr, Be, As, Ca and K and other elements were analyzed using an inductively coupled plasma mass spectrometer (ICP-MS) (Elan DRC II, PerkinElmer SCIEX, USA). Th, Zr, Be and As were determined using a standard model, argon (Ar) as reaction gas with a method detection limits (MDL) of 0.009, 0.05, 0.02 and 0.006 ppb, respectively. Ca and K were determined using dynamic reaction cell (DRC) model, ammonia ($NH_3$) as a reaction gas with MDLs of 2.2 and

0.4 ppb, respectively. The relative standard deviations (RSD) were estimated to be <10% for Th and Be, and <5% for Zr, As, Ca and K from five replicates for the concentrations determined >MDLs. Dissolved organic carbon (DOC) were determined using a TOC-VCPH analyzer (Shimadzu Corporation, Japan) with a RSD of <3% estimated from three to five replicates. DOC was analyzed as non-purgeable organic carbon (NPOC) by purging acidified samples with carbon-free air to remove DIC prior to measurement. Anions including $NO_3^-$, $SO_4^{2-}$ and $Cl^-$ were measured using an ion chromatograph (Dionex ICS-2100, Thermo Scientific, USA) with a RSD < 5% for the reported values.

### Concentration–discharge and element correlation analyses
Time-series data were analyzed using routines written in the IgorPro software to create pairs of solute concentration, river discharge, precipitation or atmospheric deposition measurements that were performed on the same day. For solutes such as Na, Ca and $SO_4^{2-}$ that displayed large seasonal changes in river concentration, the time series data were first detrended by performing a linear spline fit that was subtracted from the raw data to create a differential concentration time series, denoted e.g., ΔCa, that varied around zero. The detrended correlation plots displayed Th versus ΔCa and $\Delta SO_4^{2-}$ (Fig. S5) and ΔTh versus ΔNa (Fig. S13).

### Statistical analysis
Event coincidence analysis was performed by implementing the analytical and Monte Carlo methods described by Donges et al. (2016)[19]. Each time-series data set was converted into a binary sequence describing daily event statistics based on selected threshold.

The analysis tested the null hypothesis that the frequency of thorium excursions being preceded by either precipitation or seismic events is consistent with all events being random and uncorrelated phenomena. Donges et al.[19] derive an analytical expression for the *p*-value for observing *K* such coincidences where an event from series A precedes an event from series B by the period ΔT over a sampling period of length T.

$$P(K \geq K_{obs}) = \sum_{K' = K_{obs}}^{N_B} P(K'; N_B), \quad (1)$$

where

$$P(K; N_B) = \binom{N_B}{K} \left(1 - \left(1 - \frac{\Delta T}{T}\right)^{N_A}\right)^K \left(\left(1 - \frac{\Delta T}{T}\right)^{N_A}\right)^{N_B - K}, \quad (2)$$

and $N_A$ is the number of series A events (e.g., precipitation or seismic motion) and $N_B$ is the number of series B events (thorium excursions).

The *p*-value gives the probability of obtaining the observed number of coincidences or of a greater number of coincidences than expected if the null hypothesis (random events) is true. The smaller the *p*-value, the more likely the time series are to have a causal relationship. In this work the smallest *p*-value observed ($p = 0.12$) is not below the accepted threshold for statistical significance because there is a 12% chance that the observations were a consequence of random and uncorrelated processes.

Fig. S11 shows an example ECA analysis using an event detection threshold for river thorium increase of at least 0.1 μM in successive measurements and a seismic event threshold of ground velocity exceeding 10 μm/s. Precipitation were binned into events defined as one or more consecutive days with a maximum precipitation >2 mm/day. The thorium data for East River and Coal Creek were combined into a single sequence.

ECA analyses shown in Fig. 2 were performed parametrically, varying the thresholds for attributing seismic and precipitation events

and varying the magnitude of the time window for which A events preceding B events are counted as coincidences. These parametric ECA analyses displayed as two-dimensional heat maps of *p*-value.

This analytical expression becomes inaccurate unless there are relatively sparse events, and so the *p*-values were also evaluated using a Monte Carlo procedure. For the ECA example given in Fig. S11, the two calculations are in good agreement (Table S4).

**Laboratory leaching study**
Simulated river water was prepared to match representative geochemical conditions for groundwater sampled at the PLM sites, as reported by Tokunaga et al. (2019)[28] and Wan et al. (2019)[23]. The aqueous solution contained 10 mM NaCl and 1 mM each of $NaSO_4$ and $NaNO_3$ in MilliQ water. The sulfate and nitrate anions are generated by bedrock weathering and are potential aqueous complexing ions for the $Th^{4+}$ ion. Anaerobic solutions were prepared by first degassing by boiling, and sparged for about 1 h in nitrogen. The pH was adjusted by small additions of concentrated HCl.

Shale cores were pulverized in a mill and $2 \pm 0.1$ g portions were placed in 50-mL of aerobic or anaerobic solutions and rotated for 48 h. The suspensions were centrifuged, filtered and dissolved elemental concentrations were measured by ICP-MS.

**Water chemistry data.** River concentrations of Th, As and Zn from East River and Coal Creek, CO, and discharge data from East River, which are displayed in Figs. 1, 3 and 4, are provided in Excel format in the Source Data file. Groundwater chemistry data from floodplain meanders plotted in Fig. 3 are available from ESS-DIVE[41]. Groundwater chemistry data from hillslope wells plotted in Fig. 3 are available from ESS-DIVE[42].

Meteorological data, including precipitation, temperature and wind speed, were obtained from the meteorological station closes to the study site (KCOMKRET) and are repackaged as a gap-filled version in ESS-DIVE[43]. This station is located at Lat: 38.9150, Lon: −106.9589, Elevation: 2923 m.

Precipitation data was also analyzed from a second meteorological station, the USDA SNOTEL site NRCS-380: snotel_butte. This station is located at: Lat: 38.8943, Lon: −106.9530, Elevation: 3103 m. The data were obtained from: https://wcc.sc.egov.usda.gov/nwcc/site?sitenum=380.

Atmospheric deposition of mercury data at the Colorado Site CO97 were obtained from: http://nadp.slh.wisc.edu/data/sites/siteDetails.aspx?net=MDN&id=CO97.

Solute data from Krycklan, Sweden, were downloaded from: https://data.krycklan.se/.

Solute data from Kervidy-Naizin, France, were downloaded from: http://www7.inra.fr/ore_agrhys_eng/.

**Seismicity and GPS data.** Seismicity data for Snowmass, CO (SMCO), Sand Dunes, CO (SDCO) Mesa Verde, CO (MVCO) and Idaho Springs, CO (ISCO) covering the study period were downloaded from IRIS as binary files and converted to text using SAC. Seismicity data in the main text were converted from counts to velocity using the scale factors reported in the station metadata, which is $4.624294796 \times 10^8$ m/s for SMCO, the station used for statistical analyses.

Catalogued seismic events were identified using the Event Lookup function in *SeismicCanvas* software: https://seiscode.iris.washington.edu/projects/seismiccanvas.

GPS data were downloaded from UNAVCO: https://www.unavco.org/data/gps-gnss/data-access-methods/dai1/perm_sta.php.

## Data availability
The experiment data that support the findings of this study are available in the Source Data file or are available from the online repositories listed below. Source data are provided with this paper.

## Code availability
The commercial GIS software ArcGIS (Esri, Inc.) was used to generate the map of Fig. 1A using the basemap provided with the software (Sources: Bureau of Land Management, Esri, HERE, Garmin, USGS, NGA, EPA, USDA, Esri, HERE, Garmin, Safegraph, METI/NASA, USGS, Bureau of Land Management, EPA, NPS, USDA). The hydrologic data was provided by the National Hydrography Dataset (ver. USGS National Hydrography Dataset Best Resolution (NHD) for Hydrologic Unit (HU) 4 − 2001 (published 20191002)), accessed October 23, 2019 at URL https://www.usgs.gov/national-hydrography/access-national-hydrography-products. Geologic information was obtained from the Gunnison County Geologic Map: Streufert, Randall K., Wynn Eakins, H. Thomas Hemborg, and Matthew L. Morgan. "RS-37 Geology and Mineral Resources of Gunnison County." Geology and Mineral Resources. Resource Series. Denver, CO: Colorado Geological Survey, Division of Minerals and Geology, Department of Natural Resources, 1999. https://coloradogeologicalsurvey.org/publications/geology-mineral-resources-gunnison-colorado/.

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

## Acknowledgements

This work was conducted as part of the Watershed Function Scientific Focus Area at Lawrence Berkeley National Laboratory and was supported by the US Department of Energy (DOE) Subsurface Biogeochemical Research Program, DOE Office of Science, Office of Biological and Environmental Research, under contract no. DE-AC02 – 05CH11231. The authors thank Jiamin Wan, Tetsu Tokunaga, Caitlin Bernier and the Rocky Mountain Biological Laboratory (RMBL).

## Author contributions

B.G.: Conception of work, data analysis, drafting manuscript S.C.: Acquisition of data, revision of manuscript W.D.: Analysis and interpretation of data B.A.: Interpretation of data, revision of manuscript C.J.W.: Acquisition of data P.F.: Acquisition of data, revision of manuscript P.S.M.: Interpretation of data, revision of manuscript K.H.W.: Conception of work, substantial revision of manuscript.

## Competing interests

The authors declare no competing interests.
