## [Peer Review File · Nature Communications]

River thorium concentrations can record bedrock fracture processes including some triggered by distant seismic eventsREVIEWER COMMENTS

Reviewer #1 (Remarks to the Author):

This is an excellent paper which should definitely be published. The authors perform a careful analysis of possible causes of Thorium anomalies in river water at 2 sites in Colorado. They base their work on a rich dataset with 1 (2) year(s) of regular measurements. They successively rule out other causes than seismicity and conclude that bedrock fracturing triggered by distant seismicity is likely coupled with Thorium anomalies.

While I agree with the authors conclusions, I would like to see a couple of improvements which I do feel would strengthen the paper:

- The statistical method used is central to the conclusion of the study and should be described in the main text. I understand that this description must be brief, but I do feel that it is vital for the reader.
- As far as I can see the p-value obtained for the statistical method used depends on arbitrary selection of a threshold value (e.g. ground velocity >10 m/s) and bin sizes. Because these choices will affect the p-value obtained, a sensitivity analysis is needed to demonstrate that the conclusion is robust and not dependent on "cherry picking" of thresholds and bin sizes. I am sure that this is not the case, but it does need to be shown.
- While it is clear how correlations with rainfall and seismicity were analyzed statistically, I am unsure how other correlations (e.g. dust-on-snow events, wind speeds, major elements released by shales during weathering) were ruled out statistically.
- Fig. S8 is great. Can it be included with the main text and perhaps combined with some visual representation of the results of the experiment on river water and Thorium release, which is critical for your study.

Reviewer #2 (Remarks to the Author):

Real-time measurements of natural rock fracture in the critical zone (upper hundreds of meters of the crust) are sorely lacking and those that exist result from a narrow range of 'one time' or limited spatial scale geophysical methods. Thus, this paper's illustration and introduction of a novel geochemical technique to possibly identify periods of subsurface rock fracture represents a very important huge step forward in introducing an independent and relatively simple method for its documentation.

Overall, the paper represents an important contribution, with a strong dataset and analysis. I think the paper could be further strengthened if the authors explore the limitations and future directions a bit more. For example:

1) how might you distinguish the opening of new fractures, versus the release or redirection of an isolated reservoir of long-term accumulated Th via new hydrologic pathways? Is that possible?

I believe distinguishing between new fracture versus hydrologic changes would have implications for using the data to determine the magnitude of fracturing. In other words, a small fracture growth could lead to a large spike in Th if that small amount of cracking suddenly incorporates a large, previously isolated reservoir of Th into the system. Is there a way around this?

2) What is the justification for the 48 hour window for examining correlations? Have there been tracer tests or are there hydrographs from some of the other sampling sites to demonstrate the timescales over which fluids migrate through the entire watersheds? Otherwise, if there were fracturing activity that was more distant either literally or hydrologically, then 48h may not be enough?

3) What are your guesses for the drivers of all of the other excursions? Presumably fracture related to non-seismic sources, however, this did not exactly come across clearly. As I read through the paper from the beginning, I found myself expecting for you to make the argument that all the excursions are due to seismic events.

It felt a bit buried that, in fact, you are recognizing/concluding that your data suggest that there are lots of other fracturing events going on (non-seismic, or possibly far-field seismic? I got confused there).

In any case, that your technique may be identifying all fracturing - regardless of if it is seismogenic or not - is a very exciting finding that seems really buried in the nuances of the manuscript. (The paragraph starting at line 173.) For example, this sentence is in the middle of this paragraph, far removed from the presentation of the data that supports it in the prior paragraph. " Surprisingly, the analysis of Fig. 3 suggests that a portion of these bedrock fracture processes are triggered by small ground motions caused by distant earthquakes."

Perhaps you could put the nitty-gritty of all the 'its not this' analysis more in the supplement - just list them in the main text and refer to the figures - to give you more room to lay out the above.

Overall, I had only some other minor suggestion of edits for clarification and organization throughout the manuscript (attached) as well as:

In the supplement methods for the statistical analysis, I am always a fan of an explicit statement helping the reader interpret calculated P-values. Ex: Thus, p-values <xxx indicate that there is a <xxx% probability that y occurred randomly and so forth.

Vertical gridlines (light grey) on the timeseries data would be helpful.

Fig 1 map: using a blue outline for the watersheds at the small rendering of the map makes identifying the waterways very difficult. Add the red and green dots to the key.

In general, the fonts in all the figures are difficult to read.

Where is the weather station with respect to the two watersheds? Provide information about the distance of KCOMKRET to the watersheds in the main manuscript. Might you be missing small convective storms?

Please see and address other comments and suggestions for edits in the manuscript pdf.

All the best to you for the future of this real-time fracturing method! Martha Cary (Missy) Eppes

Reviewer #3 (Remarks to the Author):

Review:

The manuscript presented a high frequency river chemistry data (major and trace elements, up to daily monitoring for one year) and showed that the excursions of Th concentrations in shale bedrock catchment can be related to seismic events. This finding has some high level of novelty. More specifically, the authors showed the seismic events were related to propagating of seismic waves from long distance epicenters that could cause ground motion and release of elements in fracture zone groundwater to surrounding watershed. The study is important in that it explained nicely the intrinsic link between excursions of one trace element (Th) and geomechanical process (fractures). It is a

unique feature that river chemistry (high frequency monitoring) can be used to detect fracturing and low intensity earthquake events for a given watershed. However, there are several unclear points that could use more clarification or elaboration:

- 1) If the seismic events caused the movement of fractures and release of in-situ fracture water: why only Th is detected? Not the many other elements that should be released from Shale weathering?
 - 2) Is Th truly released to the environment as soluble phase? The discussion to exclude the particle Th phase needs elaboration.
 - 3) If this geomechanical process is only important in shale bedrock area, how significant impacts the Th signatures/high frequency monitoring method can bring to understand fracturing processes in other lithology? Is this Th signature only unique in the East river watershed?
- I think some moderate revision to address the above issues would significantly improve the manuscript and make it more acceptable by Nature Communication.

Line 20: changes of water chemical or physical signatures (gas discharge, water level fluctuations) related to earthquake events have been studied for some time as possible precursors for earthquake predictions (such as line 29-30). The statement that "this study is the first chemical signature of dynamic earthquake triggering" need some elaboration or clarification.

Line 32: "near-surface bedrock everywhere is fractured": please provide a citation for this statement; also, please define "fracture" such as what is the size and spacing in this statement.

Line 62-63: It is not clear why Zn is often presented as particulates. Elements such as Be, Fe, Al, Nb, Zr are more particle related than Zn which is soluble and mobile in nature. The next statement shows possible Be excursions. The Th and Be excursions could be related to particle transport.

Line 103-104: this could suggest that the release of Th is more particle related than shale weathering related (solute). The release of particles could be resulting from geomechanical processes but need more elaboration here (landslide? Fractures?).

Line 110-111: turbidity is more related to large particles while Th could be related to fine particle (if not dissolved), so the changes of turbidity is not directly related to the Th changes. Do they exhibit similar but out of phase change patterns?

Line 110-111: by the way, can landslide signatures be captured by passive seismic monitoring methods? And hence generate the correlation between seismic signatures and Th?

Line 114-116: the particle transport of Th is not completely ruled out based on the above evidence. Also, Th is very insoluble so a mechanism is needed to explain the release of Th as soluble phase and transport away from the fracture zone as soluble phase. Such as in line 124: what are the common oxyanions and are they present in groundwater near the faults/fracture areas?

Line 167-170 and Line 180-182: this is great evidence to show that the fracturing process can cause release of elements (trace solute inputs) from fracture zone groundwater to surface water. What about other elements that were from the water-rock interaction in the fault zone, why they were not detected? Why only Th shows the only responses from a list of elements that were released from water-rock interactions?

Response to All Reviewers

We thank all reviewers for thoughtful and constructive comments and questions. Before our point-by-point replies, we wish to address some common issues.

Seismic vs aseismic thorium-releasing events

The reviewers are correct that the majority of the thorium excursions are not associated with any seismic signature and we have worked to improve our description of the observations and interpretations.

We have expanded and clarified the Abstract, in particular, around this point. We also suggest a revised title: *River thorium concentrations can record bedrock fracture processes including some triggered by distant seismic events*

Statistical analysis of potential causality between temporally separate events

Two reviewers asked for more information about the 2-day time window used in the submitted manuscript for statistical analysis of whether events preceding thorium excursions occurred more frequently than random. This initial 2-day time window was selected because the interval between river water sampling was often 2 days and because the most important time constant for thorium concentration decay varies from 1–2 days. However, we did not perform a systematic assessment at that time.

For the revised version, we adopted a parametric approach for the statistical test called event coincidence analysis (ECA), varying the thresholds selected for event identification and varying the time window. The analysis finds a region of consistently lower *p-values* for seismicity versus precipitation, in agreement with the prior manuscript. In addition, we have added parametric ECA for an independent precipitation dataset and for the wind speed data, also finding no correlation.

The lowest *p-value* is still not 'statistically significant' but the 2D plots of *p-value* versus threshold and window criteria indicate that there could be a non-random relationship between thorium and seismicity that is not suggested for thorium and precipitation.

Why thorium?

Our manuscript describes evidence that thorium can provide a chemical signature of bedrock fracture processes in the two neighboring watersheds where we are conducting research. It is not our view, however, that thorium is the only element released by these processes, *rather it is the only element detected at a level that enables distinctive concentration patterns to be revealed*. This is because the background concentration is so low that sub-ppb excursions are detectable over many days. In East River watershed, Be and As show some simultaneous concentrations excursions with Th, but these concentration patterns are less distinct. This has implications for the use of Th or other trace elements on detecting fracture processes in other watersheds, and we expand the discussion of this point.

Reviewer #1 (Remarks to the Author):

This is an excellent paper which should definitely be published. The authors perform a careful analysis of possible causes of Thorium anomalies in river water at 2 sites in Colorado. They base their work on a rich dataset with 1 (2) year(s) of regular measurements. They successively rule out other causes than seismicity and conclude that bedrock fracturing triggered by distant seismicity is likely coupled with Thorium anomalies.

While I agree with the authors conclusions, I would like to see a couple of improvement which I do feel would strengthen the paper:

- The statistical method used is central to the conclusion of the study and should be described in the main text. I understand that this description must be brief, but I do feel that it is vital for the reader.

We have added a brief description of the statistical method to the main text.

- As far as I can see the p-value obtained for the statistical method used depends on arbitrary selection of a threshold value (e.g. ground velocity >10 m/s) and bin sizes. Because these choices will affect the p-value obtained, a sensitivity analysis is needed to demonstrate that the conclusion is robust and not dependent on "cherry picking" of thresholds and bin sizes. I am sure that this is not the case, but it does need to be shown.

Thank you for prompting us to do a more complete assessment of the assumptions underlying the statistical test. We now include figures reporting parametric analysis of the Event Analysis Coincidence as a function of event thresholds and the length of the time windows. The results continue to show a weak correlation between seismicity and thorium that is not observed when comparing precipitation and thorium.

We have added a new **Figure 2** that gives the improved Event Coincident Analysis.

- While it is clear how correlations with rainfall and seismicity were analyzed statistically, I am unsure how other correlations (e.g. dust-on-snow events, wind speeds, major elements released by shales during weathering) were ruled out statistically.

We performed one Event Correlation Analysis (ECA) to test the correlation between thorium excursions and wind speed, finding no correlation for a manually chosen time windows.

For this resubmission, we have added parametric ECA for the wind speed data, which are included in **Figure 2**.

For the original submission, we did not perform ECA or other statistical approach to test correlations between Th and other solutes, ground motion or temperature. Instead, we visually inspected time-series data for each element compared with thorium and we generated thorium-element correlation plots. The time-series data and correlation plots both indicated that only Be and As were (partly) correlated with thorium. However, we accept that this is not a true statistical test.

For this resubmission, we performed a modified version of ECA for the major solutes, first detrending the major solute time-series data using a linear spline function. This did not find any support for correlation between major solutes and thorium. Because this is in agreement with expectation, we have not added this analysis to the revised manuscript, but we have added new correlation plots between thorium and detrended major ion data.

We have added CQ and Correlation plots in Supplementary **Figure S5**.

We could not find a relevant data set for dust-on-snow events in the greater East River watershed. However, the thorium events do not match the seasonality of eolian transport and deposition.

- Fig. S8 is great. Can it be included with the main text and perhaps combined with some visual representation of the results of the experiment on river water and Thorium release, which is critical for your study.

It is our preference to keep Fig. S8 and the experimental results in the Supplement for two reasons. First, these data support the hypothesis that thorium excursions are the result of fracturing that exposes fresh shale rock to ground water. However, this is not the only plausible origin of thorium. As Reviewer 2 writes: thorium signatures could be caused “opening of new fractures” or “the release or redirection of an isolated reservoir of long-term accumulated Th via new hydrologic pathways.”

Second, we are presently writing a manuscript that will describe the pathway of shale weathering at this site, and would like to include a version of the tomographic analysis in that paper.

Reviewer #2 (Remarks to the Author):

Real-time measurements of natural rock fracture in the critical zone (upper hundreds of meters of the crust) are sorely lacking and those that exist result from a narrow range of ‘one time’ or limited spatial scale geophysical methods. Thus, this paper’s illustration and introduction of a novel geochemical technique to possibly identify periods of subsurface rock fracture represents a very important huge step forward in introducing an independent and relatively simple method for its documentation.

Overall, the paper represents an important contribution, with a strong dataset and analysis. I think the paper could be further strengthened if the authors explore the limitations and future directions a bit more. For example:

1) how might you distinguish the opening of new fractures, versus the release or redirection of an isolated reservoir of long-term accumulated Th via new hydrologic pathways? Is that possible?

I believe distinguishing between new fracture versus hydrologic changes would have implications for using the data to determine the magnitude of fracturing. In other words, a small fracture growth could lead to a large spike in Th if that small amount of cracking suddenly incorporates a large, previously isolated reservoir of Th into the system. Is there a way around this?

The Reviewer pinpoints a question that we were not able to resolve during our analysis of the data. We tentatively expect that it will be possible to perform subsurface hydrologic simulations that predict river water thorium signatures based on different fracture processes, using realistic descriptions of bedrock fracture permeability. Colleagues in the larger project are developing models for water storage and transport through soil, regolith and fractured bedrock (Sprenger, Carroll) and we plan to develop this direction in future work. However, the numerical modeling required is out of scope of this paper.

We added this statement at Line 199.

The above observations provide evidence that abrupt bedrock fracture processes can generate detectable trace solute inputs to ground and stream water. These could include processes that create new water-rock interactions, such as fracture creation or slip, as well as processes that change subsurface hydraulic pathways

2) What is the justification for the 48 hour window for examining correlations? Have there been tracer tests or are there hydrographs from some of the other sampling sites to demonstrate the timescales over which fluids migrate through the entire watersheds? Otherwise, if there were

fracturing activity that was more distant either literally or hydrologically, then 48h may not be enough?

As described above, now include a parametric calculation of the effect of time-window size. We find that the weak correlation between seismicity and thorium observed for a time window of 24 – 48 hours is not observed at greater time windows.

We have added a new **Figure 2** that gives the parametric Event Coincident Analyses.

3) What are your guesses for the drivers of all of the other excursions? Presumably fracture related to non-seismic sources, however, this did not exactly come across clearly. As I read through the paper from the beginning, I found myself expecting for you to make the argument that all the excursions are due to seismic events.

It felt a bit buried that, in fact, you are recognizing/concluding that your data suggest that there are lots of other fracturing events going on (non-seismic, or possibly far-field seismic? I got confused there).

In any case, that your technique may be identifying all fracturing - regardless of if it is seismogenic or not - is a very exciting finding that seems really buried in the nuances of the manuscript. (The paragraph starting at line 173.) For example, this sentence is in the middle of this paragraph, far removed from the presentation of the data that supports it in the prior paragraph. " Surprisingly, the analysis of Fig. 3 suggests that a portion of these bedrock fracture processes are triggered by small ground motions caused by distant earthquakes."

Perhaps you could put the nitty-gritty of all the 'its not this' analysis more in the supplement - just list them in the main text and refer to the figures - to give you more room to lay out the above.

We thank the reviewer for highlighting this important point. In agreement with the Reviewers, we infer that most thorium excursions are caused by watershed fracture processes that have no local or distant seismic signature – that is, they are aseismic. This important point was not clearly made in the original manuscript.

We do not know the drivers of the aseismic thorium excursions. However, we expect that they are caused by slower processes that alter the stress field in the watershed, including variation in groundwater loading, tectonic forces, and the annual variation in surface elevation caused mainly by seasonal variation in surface temperatures (data that were already presented in what is now **Fig. S3**).

We have revised the Abstract so that this point is more clearly explained.

We have revised sections of the text to emphasize that, while we consider all thorium excursions to be associated with fracture processes, only a minority (up to 40%) are associated with seismic motion.

Overall, I had only some other minor suggestion of edits for clarification and organization throughout the manuscript (attached) as well as:

In the supplement methods for the statistical analysis, I am always a fan of an explicit statement helping the reader interpret calculated P-values. Ex: Thus, p-values <xxx indicate that there is a <xxx% probability that y occurred randomly and so forth.

We added the following explanation to the relevant section of the Supplement:

The p -value gives the probability of obtaining the observed number of coincidences or of a greater number of coincidences than expected if the null hypothesis (random events) is true. The smaller the p -value, the more likely the time series are to have a causal relationship. In this work the smallest p -value observed ($p = 0.12$) is not below the accepted threshold for statistical significance because there is a 12% chance that the observations were a consequence of random and uncorrelated processes.

Vertical gridlines (light grey) on the timeseries data would be helpful.

We understand the motivation for the reviewer's request, which is to help the reader line up coincident or preceding events between the stacked time-series data. In response, we drafted a number of revised figures incorporating gridlines. However, we feel that the gridlines mainly make these data-rich plots more cluttered and do not significantly aid the reader. Specifically, virtually none of the gridlines end up being in the right place for helpful comparisons. So we respectfully prefer to keep the plots without gridlines.

Fig 1 map: using a blue outline for the watersheds at the small rendering of the map makes identifying the waterways very difficult. Add the red and green dots to the key.

The watershed outline has been thickened, the waterways rendered more clearly, and the field sampling locations (red and green dots) and weather stations added to the legend.

In general, the fonts in all the figures are difficult to read.

We have re-prepared all figures in the Main Text and almost all figures in the Supplement. Throughout, we have increased text size and increased legibility.

Where is the weather station with respect to the two watersheds? Provide information about the distance of KCOMKRET to the watersheds in the main manuscript. Might you be missing small convective storms?

The KCOMCRE2 weather station is 1.09 km from the principal study site, the East River Pump House, and at a similar elevation. Although we frequently witnessed small and sudden summer thunderstorms in the East River valley, we anticipate that most of these are captured by this weathering station.

We looked for another weather station closet to the Coal Creek sampling site but could not find any station in that watershed operating 2016-217. The USDA SNOTEL site is located on Mount Crested Butte, about 200 m higher and about 3 km closer to Coal Creek. We performed the same Event Coincident Analysis (ECA) for the SNOTEL precipitation and did not find any greater correlation between precipitation and thorium excursions.

We have added the KCOMCRET and SNOTEL weather stations to **Figure 1**.

We have added parametric ECA analysis for the SNOTEL precipitation data to the new **Figure 2**.

Please see and address other comments and suggestions for edits in the manuscript pdf.

We have incorporated responses to all comments in the manuscript PDF file.

All the best to you for the future of this real-time fracturing method! Martha Cary (Missy) Eppes

Reviewer #3 (Remarks to the Author):

Review:

The manuscript presented a high frequency river chemistry data (major and trace elements, up to daily monitoring for one year) and showed that the excursions of Th concentrations in shale bedrock catchment can be related to seismic events. This finding has some high level of novelty. More specifically, the authors showed the seismic events were related to propagating of seismic waves from long distance epicenters that could cause ground motion and release of elements in fracture zone groundwater to surrounding watershed. The study is important in that it explained nicely the intrinsic link between excursions of one trace element (Th) and geomechanical process (fractures). It is a unique feature that river chemistry (high frequency monitoring) can be used to detect fracturing and low intensity earthquake events for a given watershed. However, there are several unclear points that could use more clarification or elaboration:

1) If the seismic events caused the movement of fractures and release of in-situ fracture water: why only Th is detected? Not the many other elements that should be released from Shale weathering?

Thank you for this question. We agree with the reviewer that the fracture processes that we infer are responsible for introducing thorium into the watersheds must also release many other elements. Thorium, however, is the only element that shows the distinctive concentration patterns in both watershed.

We conclude that the detection of fracture processes through element concentration changes requires that concentration change to be detectable above the background. This behavior likely requires either (1) that the element inputs into the watershed are very low or (2) that there are processes in the watershed that keep the background concentrations of the element extremely low.

The first scenario is likely illustrated by As and Be. In the East River watershed, beryllium and arsenic show similar concentration excursions as thorium, while in the Coal Creek watershed, Be and As are not correlated with Th. This is likely because Coal Creek is impacted by sulfidic rock weathering that continuously releases metals leading to a continuously higher background.

The second scenario may explain the Th data. For thorium, the low intrinsic solubility of this 4+ cation, which the reviewer mentions, could be such a mechanism. It is our hope to be able to pursue further studies of the geochemical speciation of thorium that enables its use as a signature of fracture processes.

This discussion is now included in the text.

2) Is Th truly released to the environment as soluble phase? The discussion to exclude the particle Th phase needs elaboration.

In response to this question, and the reviewer's other questions below, we performed more extensive analysis of East River chemistry to distinguish element concentration patterns. We compared time-series data (original **Figs. 1 and S2**), concentration-discharge (usually termed C-Q) analyses (new **Fig. S5a**), and correlations between element concentrations (new **Fig. S5b**) for selected elements. In response to the Reviewer's comment (below), we now compare Th, Be and As with highly insoluble Zr rather than Zn.

The CQ plots for Zr, Th, and Be (new **Fig. S5a**) all have similar patterns that could indicate particulate transport: the concentration is typically very low or below detection, regardless of discharge; and there is no correlation between non-zero concentration values and discharge. The CQ plot for As is similar, except that there is a constant (“chemostatic”) background at all discharge values. These CQ trends are strikingly different from soluble species (sulfate and organic carbon). Conventional interpretation of the CQ data, therefore, suggests Zr, Th and Be exhibit particulate behavior, while As is present as both soluble species and particulates.

The correlation plots for Th-Be, Th-As and Th-Zr are shown in new **Fig. S5b**. The plots show that Th is mostly uncorrelated with Th and Be, except for a subset of dates in both cases where a cluster of Th-Be and Th-As points are shown. In contrast, Th is completely uncorrelated with Zr. This analysis could indicate that Th, Be and As are occasionally present in the same particulate phase that does not contain Zr.

In summary, we agree with the reviewer that our data and interpretation cannot confidently establish whether thorium is present as dissolved solute or in a fine particulate phase. Moreover, conventional analysis of the element concentration patterns would suggest that Th is present in particulates.

However, the time-series data showing the thorium concentration profiles suggest that conventional analysis is not appropriate in this case. We consider that the concentration decay curves of Th may also be due to mixing and dilution of a soluble form of thorium.

We have added CQ and Correlation plots in Supplementary **Figure S5**.

We have expanded the discussion of the form of Th (dissolved vs particulate).

3) If this geomechanical process is only important in shale bedrock area, how significant impacts the Th signatures/high frequency monitoring method can bring to understand fracturing processes in other lithology? Is this Th signature only unique in the East river watershed?

In our original submission, we sought to answer this question by searching for time-series river chemistry data from other watersheds with long-term monitoring. As shown in Supplemental **Figure S12** we only found two relevant date sets and, of the two, only the data from Kervidy-Naizin, France, included high-frequency (daily) water concentrations.

At Kervidy-Naizin, the thorium data are clearly correlated with river discharge and with major solute concentrations (e.g., Na). This is in complete contrast with East River data, where the thorium data exhibit no obvious correlations between discharge and major solutes. We conclude that thorium concentrations at the Kervidy-Naizin watershed do not show any influence of fracture processes. Thus, the thorium data at East River are unique to date.

We tentatively speculate:

1. that the influence of fracture processes on river chemistry may be stronger in mountainous watersheds than in lower-elevation catchments. In mountainous watershed, the depth to bedrock is typically smaller and a greater fraction of subsurface permeability is associated with bedrock fracture networks rather than overlying soil.
2. We do not have clear evidence for bedrock lithology being important. While East River is dominantly situated on Mancos Shale, which we showed can release Th, the Coal Creek watershed that also exhibits strong Th excursions is dominantly situated on granite.

We have added a new Supplementary **Figure S13** showing correlations plots between [Th] and [Na] at East River and Kervidy-Naizin. At Kervidy-Naizin, a weak correlation is observed while at East River no correlation is found.

We have expanded the discussion as to whether Th (or other elements) might be sensitive to fracture processes in other locations.

I think some moderate revision to address the above issues would significantly improve the manuscript and make it more acceptable by Nature Communication.

Line 20: changes of water chemical or physical signatures (gas discharge, water level fluctuations) related to earthquake events have been studied for some time as possible precursors for earthquake predictions (such as line 29-30). The statement that “this study is the first chemical signature of dynamic earthquake triggering” need some elaboration or clarification.

We have sought to clarify this point. We already cite (lines 29-30) important papers that have shown both geochemical precursors to earthquakes and geochemical (and hydrologic) consequences of fault motion. However, none of these papers (and no others that we are aware of) have linked *local* geochemical changes to *distant* earthquakes. That finding is the basis for the conclusion that fault or fracture motions in Colorado could be triggered by earthquakes in Nevada or California.

Line 32: “near-surface bedrock everywhere is fractured”: please provide a citation for this statement; also, please define “fracture” such as what is the size and spacing in this statement.

We have deleted this overly expansive statement. We simply conclude the first paragraph of the Introduction with the (unchanged) statement.

“However, there are few methods and studies that detect any changes to near-surface bedrock fracture.”

Line 62-63: It is not clear why Zn is often presented as particulates. Elements such as Be, Fe, Al, Nb, Zr are more particle related than Zn which is soluble and mobile in nature. The next statement shows possible Be excursions. The Th and Be excursions could be related to particle transport.

We thank the reviewer for this point. In our dataset, a number of elements including Zn exhibited time-series behavior that could indicate transport in colloids. However, the reviewer is correct that this element is not usually associated with colloidal transport. We now refer to the paper by Olschansky et al., (2018), working at the Catalina-Jemez Critical Zone Observatory, that both separated and measured the colloidal fraction and performed Principal Component Analysis on stream water data. The results showed that Fe, Zr, Mn, Ti and Al were present in particulate phase and that particulate transport was associated with very high variance in concentration (i.e., large and sudden variations in concentration in time series data). Our Th data are not correlated with any of these elements.

We have expanded the discussion about particulate transport.

We now compare the Th data with Zr and not Zn.

Line 103-104: this could suggest that the release of Th is more particle related than shale weathering related (solute). The release of particles could be resulting from geomechanical processes but need more elaboration here (landslide? Fractures?).

Line 110-111: turbidity is more related to large particles while Th could be related to fine particle (if not dissolved), so the changes of turbidity is not directly related to the Th changes. Do they exhibit similar but out of phase change patterns?

Line 110-111: by the way, can landslide signatures be captured by passive seismic monitoring methods? And hence generate the correlation between seismic signatures and Th?

Landslides can be detectable by passive seismic monitoring, but detecting the relatively small events that have been witnessed at East River will require a network of sensors in the watershed, which does not presently exist. It is our hope that the present study will motivate investment into a passive seismic sensing network in this field site to more precisely relate surface and subsurface motions with river geochemistry.

We added a sentence and the reference below about passive seismic monitoring.

M. Le Breton, N. Bontemps, A. Guillemot, L. Baillet and É. Larose (1990) Landslide monitoring using seismic ambient noise correlation: challenges and applications. *Earth-Science Reviews* 216, 103518.

Line 114-116: the particle transport of Th is not completely ruled out based on the above evidence. Also, Th is very insoluble so a mechanism is needed to explain the release of Th as soluble phase and transport away from the fracture zone as soluble phase. Such as in line 124: what are the common oxyanions and are they present in groundwater near the faults/fracture areas?

We agree that Th⁴⁺ is highly insoluble in water but solubility can be increased by complexation with oxyanions that are present in groundwater at these watersheds. In a paper that we already cited, Langmuir and Herman (1980) showed that sulphate is the most common inorganic complexing oxyanion. As a consequence of pyrite oxidation in weathering shale, the groundwater and stream waters in both the East River and Coal Creek watersheds contain at least 10⁵ times more sulfate than thorium. With a complexation constant $K \sim 10$, sulfate alone could solubilize thorium. In addition, phosphate, nitrate, chloride and fluoride are present and likely additional complexing inorganic ions.

Organic oxyanions such as oxalate have still higher affinity for Th⁴⁺. Although we have not analyzed the molecular composition of organic compounds in the watersheds, the organic carbon contents of both East River and Coal Creek vary from about 10 – 300 nM, which is 1 to 3 orders of magnitude greater than Th.

Langmuir, D. and Herman, J. S. (1990) The mobility of thorium in natural waters at low temperatures. *GCA* 44, 1753-1766.

Line 167-170 and Line 180-182: this is great evidence to show that the fracturing process can cause release of elements (trace solute inputs) from fracture zone groundwater to surface water. What about other elements that were from the water-rock interaction in the fault zone, why they were not detected? Why only Th shows the only responses from a list of elements that were released from water-rock interactions?

We have given an answer to this question at the beginning of this response.

REVIEWERS' COMMENTS

Reviewer #1 (Remarks to the Author):

This is a re-review and I am fully satisfied that the authors have made an thorough and careful attempt to address all of my concerns. I recommend publication of this manuscript in its present form.

Reviewer #3 (Remarks to the Author):

I have read the responses to reviewer#3 and the revision and think all of my previous comments and suggestions have been adequately addressed by the revision. the revision also improved significantly in its clarity for linking the release of Th and other possible trace elements with fracturing events and the possible mechanisms. I have no further comments or issues and am happy to recommend publication of the revision. I would like to thank the authors for their time and effort making the changes and clarifications.